# Heinrich events show two-stage climate response in transient glacial simulations

Florian Andreas Ziemen[1], Marie-Luise Kapsch[1], Marlene Klockmann[1], and Uwe Mikolajewicz[1]

[1]Max Planck Institute for Meteorology, Bundesstraße 53, 20146 Hamburg, Germany

**Correspondence:** Florian A. Ziemen (florian.ziemen@mpimet.mpg.de)

**Abstract.** Heinrich events are among the dominant modes of glacial climate variability. During these events, massive iceberg armadas were released by the Laurentide Ice Sheet, sailed across the Atlantic where they melted and released fresh water, as well as detritus that formed characteristic layers on the sea floor. Heinrich events are known for cold climates in the North Atlantic region and global climate changes. We study these events in a fully coupled complex ice sheet–climate model with synchronous coupling between ice sheets and oceans. The ice discharges occur as internal variability of the model with a recurrence period of 5 kyr, an event duration of 1–1.5 kyr, and a peak discharge rate of about 50 mSv, roughly consistent with reconstructions. The climate response shows a two-stage behavior, with freshwater release effects dominating the surge phase and ice-sheet elevation effects dominating the post-surge phase. As a direct response to the freshwater discharge during the surge phase, the deepwater formation in the North Atlantic decreases and the North Atlantic deepwater cell weakens by 3.5 Sv. With the reduced oceanic heat transport, the surface temperatures across the North Atlantic decrease, and the associated reduction in evaporation causes a drying in Europe. The ice discharge lowers the surface elevation in the Hudson Bay area and thus leads to increased precipitation and accelerated ice sheet regrowth in the post-surge phase. Furthermore, the jet stream widens to the north, which contributes to a weakening of the subpolar gyre, and a continued cooling over Europe even after the ice discharge. This two-stage behavior can explain previously contradicting model results and understandings of Heinrich Events.

## 1 Introduction

Heinrich events (Heinrich, 1988), one of the two dominant modes of glacial climate variability, are associated with a reorganization of the North Atlantic Ocean circulation (see Clement and Peterson, 2008, for a review). They are quasi-periodic with a recurrence period of about 7 kyr and fall into the Dansgaard-Oeschger cold periods (*stadials*). In Heinrich events, iceberg armadas released from the Laurentide Ice Sheet spread a characteristic trace of detritus from the Hudson Strait area across the North Atlantic seafloor. In the traditional view, the climate changes during Heinrich events that go beyond a normal Dansgaard-Oeschger stadial result from the effect of the freshwater released by the melting icebergs on the ocean circulation. The importance of the freshwater effect was put in question by the first study with a coupled ice sheet–General Circulation Model (GCM) system (Roberts et al., 2014b). In this study the freshwater effect is minor. Instead, the climate changes stem

from ice sheet-atmosphere interactions. We present the second coupled ice sheet–GCM study of Heinrich events and show that the climate response in our model is a succession of both effects.

The periodic iceberg release in Heinrich events can be explained by a periodic acceleration (*surging*) of the ice flow in the Hudson Strait resulting from buildup–collapse (*binge-purge*) cycles of the Laurentide Ice Sheet (MacAyeal, 1993). In the multi-millennial buildup phase, the ice sheet grows until it becomes unstable and then surges into the ocean. This view has been put into question because Heinrich events always fall into Dansgaard-Oeschger stadials, and so a triggering (setting the timing) or causation (providing a mechanism/reason) by sub-surface ocean melt has been suggested (Moros et al., 2002; Hulbe et al., 2004; Álvarez-Solas et al., 2011; Bassis et al., 2017). In the first three-dimensional ice sheet model study of these cycles, the timing of self-sustained surge cycles could be phase-locked to a 1500-year cycle resembling the Dansgaard-Oeschger cycles by perturbing the ice sheet at the mouth of Hudson Strait (Calov et al., 2002), indicating the possibility of a self-sustained oscillation where the individual events are triggered by ocean processes. While idealized model studies have shown the surge-behavior to be highly model and parameter dependent (Calov et al., 2010), they are a robust feature of the ice sheet model used in this study (Van Pelt and Oerlemans, 2012; Feldmann and Levermann, 2017). As our setup does not exhibit Dansgaard-Oeschger cycles, the study of the relationship between Heinrich events and Dansgaard-Oeschger cycles in coupled ice sheet–climate models will remain open for future studies.

In the traditional view, the freshwater released from the melting icebergs stabilizes the ocean stratification in the North Atlantic, this reduces the deepwater formation and thus the ocean heat release, thereby causing large-scale climate changes. This view is supported by freshwater perturbation (*hosing*) experiments using climate models (e. g. Maier-Reimer and Mikolajewicz, 1989; Schiller et al., 1997; Stouffer et al., 2006; Kageyama et al., 2013). These experiments can explain the large-scale cooling and the drying in Eurasia that are commonly observed in proxies.

In the coupled ice sheet–GCM study of Roberts et al. (2014b), the topography was the most important factor for determining the climate response and outweighed effect of the freshwater release (0.05 Sv) in the ocean. The surge in the Hudson strait draws down the surface elevation in the Hudson Bay area. This opens space for changes in the atmospheric circulation that affect the climate on a large scale. Only in a sensitivity experiment, where Roberts et al. (2014b) multiplied the freshwater signal from the ice sheets by a factor of 20, both topography and freshwater reached equal importance. However, the authors used an 1:100 asynchronous coupling between the climate model and the ice sheet model. Hence, while a Heinrich event lasted 1-2 kyr in the ice sheet model, it only lasted 10-20 years for the climate model. Hosing experiments have shown that the ocean needs several decades to fully respond to a freshwater forcing (e. g. Stouffer et al., 2006; Otto-Bliesner and Brady, 2010). Therefore, the asynchronous coupling likely suppressed a part of the ocean response. Furthermore, the climate model FAMOUS employed by Roberts et al. (2014b) is rather insensitive to freshwater forcing in the Labrador Sea where much of the freshwater in Roberts et al. (2014b) is released (Smith and Gregory, 2009).

Here, we present the first results from coupled ice sheet–GCM Heinrich event simulations with synchronous coupling between the ice sheets and the ocean. Thus, the full response of the ocean circulation can unfold and we can study the different phases of the climate response. With this setup, we investigate the characteristic properties of Heinrich events in general. To obtain an archetypal event, we perform a composite analysis of four Heinrich events obtained from three transient experiments.

We first describe the setup and the experiments, then detail the two-stage structure we observe in the composite and finally discuss how this unifies the results of freshwater hosing experiments with those of Roberts et al. (2014b).

## 2 Model and setup

### 2.1 Model and coupling

The simulations were performed with a coarse resolution (T31/GR30) setup of the CMIP3 Max Planck Institute climate model ECHAM5/MPIOM/LPJ (Mikolajewicz et al., 2007b) coupled with a 20 km Northern Hemisphere setup of the modified Parallel Ice Sheet Model (mPISM) version 0.3 (Bueler and Brown, 2009; Winkelmann et al., 2011 for PISM). In this setup, the models are coupled bidirectionally without flux correction or other anomaly methods. mPISM and a similar setup are described in Ziemen et al. (2014). The main characteristics of the setup, and the differences to Ziemen et al. (2014) are outlined in the
following.

In all simulations used for the analysis, a configuration was used where atmosphere and ocean are coupled with a periodic-synchronous 1:10 coupling (Voss and Sausen, 1996; Mikolajewicz et al., 2007a). The motivation for this is that the atmosphere has no long-term memory (this resides in ocean and ice sheets), but consumes more than 90% of the CPU time of the coupled model system. Periodic-synchronous coupling means that the atmosphere model is only run for one out of ten years. This
reduces the computational expense drastically and speeds up the simulations by a factor of three (wallclock time). After each fully coupled year, the ocean is forced for nine years with atmospheric fields, that are obtained by cycling through the previous five fully coupled years. These settings turned out to be a good compromise between having a sufficiently large archive of atmospheric forcing to adequately represent the inter-annual variability of the atmospheric forcing and minimizing the delay in the forcing from a large archive. A too small archive leads to a large model drift in the ocean-only phases and thus corrupts
the climate of the coupled model, a large archive introduces a large delay. Whereas this coupling technique has only minor effect on the simulated climate response to long-term changes, this technique should not be used for short-term changes or the analysis of short-term variability. With the settings used here, the periodic-synchronous coupling introduces a lag of up to 50 years in the atmosphere-ocean system. The average lag of about 25 years is less than 10% of the 300-year averaging window used throughout most of the analysis. Furthermore, most of the changes in the ice sheets driving the atmosphere-ocean system
occur on a longer time scale, so the lag can be neglected in the analysis. Parallel to the climate model, the ice sheet model is run for ten years, so ice sheets and ocean are on the same time scale. In the analysis of atmosphere and ocean fields, only the fully coupled years are used. These are also used to obtain the fields for forcing the ice sheet model. Thus, when we speak of a 300 year mean of the ocean 2D fields, only 30 years of data with a 10-year spacing are used. This should yield comparable results to the full data as long as there are no signals with periods of exact multiples of 10 years involved (aliasing).
The surface mass balance is computed using downscaled precipitation and temperatures in a Positive Degree Day (PDD, Reeh, 1991) scheme in the ice sheet model. The PDD scheme employs the Calov–Greve integral method (Calov and Greve, 2005) to compute PDDs from monthly mean temperatures and standard deviations. The PDDs are then converted to snow and ice melt. As in Ziemen et al. (2014), the temperature standard deviation for the PDD scheme is computed from 6-hourly atmo-

sphere model output. Extending this method, a minimum sub-monthly standard deviation of 4 K is prescribed. This prevents the standard deviation from falling too low in areas, where ECHAM5 simulates melt and limits the surface temperature to 0°C.

Shelf basal melt is computed from ocean temperature and salinity using the three-equation model of Holland and Jenkins (1999). In contrast to Ziemen et al. (2014), iceberg calving and shelf basal melt are limited to grid cells that are connected to the open ocean at least via a channel with a minimum width of one grid cell and a minimum depth of 400 m. Such channels may be covered by a floating ice shelf.

In the ice sheet model, the surge behavior is strongly influenced by the basal friction law for the sliding regions, that relates the basal velocity $\boldsymbol{u}_b$ to the basal shear stress $\boldsymbol{\tau}_b$. Sadly, this relationship is extremely poorly constrained (e.g. Calov et al., 2002). As in Ziemen et al. (2014), we use a linear sliding law

$$\boldsymbol{u}_b = -c\boldsymbol{\tau}_b^m \tag{1}$$

with the sliding coefficient $c = 1\,\mathrm{m\,Pa^{-1}\,a^{-1}}$ and m=1. This implies that on soft, deforming, water saturated sediments the ice can slide with very little friction. Calov et al. (2002) tested values of c between 0.01 $\mathrm{m\,Pa^{-1}\,a^{-1}}$ and 0.1 $\mathrm{m\,Pa^{-1}\,a^{-1}}$, and obtained cyclic surges for values above 0.03 $\mathrm{m\,Pa^{-1}\,a^{-1}}$. They also were able to obtain surges for quadratic (m=2) and cubic (m=3) growth of the sliding speed with the basal shear stress. Similarly, Van Pelt and Oerlemans (2012) found millennial scale surges for the highest sliding coefficients (low till friction angle in their paper). Roberts et al. (2016) tested values for c ranging from 0.0005 $\mathrm{m\,Pa^{-1}\,a^{-1}}$ to 1 $\mathrm{m\,Pa^{-1}\,a^{-1}}$ and found surges for all but the lowest value.

Ice sheet elevation, fresh water fluxes, and ice-ocean heat fluxes are fed back to the climate model. As another improvement over the setup of Ziemen et al. (2014), the river routing directions in the hydrology scheme are computed interactively to account for ice sheet changes and isostatic rebound. The direct ice discharge into the ocean is modeled as freshwater input with an associated negative heat input that turns the fresh water flux into an ice flux. If the ocean is at the freezing point, this leads to immediate sea ice formation. Under pre-industrial conditions, this parametrization already captures most of the climatic effects of Northern Hemisphere icebergs (Bügelmayer et al., 2015).

## 2.2 Experiments

We analyze a composite of four Heinrich events from three experiments (Fig. 1a, Tab. 1, Section 2.3 for details on the composite analysis). These are one long experiment (ExA in the following, see video supplement) starting at 42 kyr BP and two shorter experiments (ExB and ExC) starting at 23 kyr BP. From ExA, we discard the first 10 kyrs as final phase of the spin-up and obtain two events starting at 31 kyr BP (A1), and 25.7 kyr BP (A2). From ExB and ExC, we obtain two Heinrich events starting at 22.4 kyr BP: B1 and C1. Later Heinrich events show an interference of the deglaciation and are discarded. All experiments were forced with transient insolation and greenhouse gas concentrations from reconstructions (Berger, 1978; Spahni and Stocker, 2006a, b; Lüthi et al., 2008). The Antarctic Ice Sheet and the bedrock topography outside the ice sheet domain were prescribed based on the ICE-5G 21 ka topography (Peltier, 2004). The topography is corrected for a global uniform sea level change computed from the modeled ice volume. In ExA, and ExB, the dynamic river discharge scheme routes the Nile into the Red Sea during the glacial. In ExC this scheme is improved and the Nile always feeds into the Mediterranean. The technical differences

between the three experiments are small enough to consider them as comparable. We chose these simulations as technically quasi-identical subset from various simulations that were performed when working on a model that is able to simulate the last deglaciation. As the simulations consumed considerable resources, we refrained from performing a dedicated ensemble, but made use of the available data.

The common spinup of all experiments began with a LGM steady state experiment (Ziemen et al., 2014) exhibiting a large spurious ice sheet over Siberia that connected to the Cordilleran part of the Laurentide Ice Sheet via Alaska (Fig. 8 of Ziemen et al. (2014), its re-emergence can be seen in Fig. 2). This ice sheet partly is an artifact of steady-state LGM simulations (Heinemann et al., 2014) and is common in coupled ice-climate modeling (e. g. Bauer and Ganopolski, 2017). Its growth can be reduced by modeling the effect of dust (Krinner et al., 2011). During the first phase of the spinup, the ice sheet was removed
by setting the ice thickness between 90 °E and 140 °W to zero and relaxing the bedrock in this region. The model was then equilibrated with fixed surface mass balance forcing and constant removal of all ice between 90 °E and 140 °W for 5000 years. This yielded ice sheets with smooth margins that did not extent into Eastern Siberia or Alaska. From this state, two experiments were started by setting the time to 42 kyr BP and using transient orbital and greenhouse gas forcing. One of them is ExA. The other is an asynchronously coupled test simulation, from which ExB and ExC were branched off at 23 kyr BP. While the ice
sheet in Siberia was removed in the first phase of the spin-up, in the following phase, it formed again (Fig. 2).

## 2.3   Composite analysis

The mechanisms related to the surging of the ice sheet are highly non-linear, leading to variability between individual realizations of the modeled events even under quasi-identical conditions (Soucek and Martinec, 2011). This variability is further amplified by feedbacks in the fully coupled ice sheet–climate model. To reduce the influence of variability and thus obtain
more robust results, we perform all further analysis on a composite of all four events. This creates an archetypal event that has the characteristics that are common to all events. To create the composite, we align the events on a common time scale. We base this alignment on the freshwater discharge at the mouth of Hudson Strait to extract the peak discharge phase. The duration of the start phase of the surge varies from event to event, so aligning for the same start time of the surge would smear out the peak discharge. However, the rising flank of the discharge signal is steep in all events (Fig. 1b), and closely precedes
the peak discharge. Hence, all events are aligned so they cross 30 mSv Hudson Strait discharge in year 250, and the first surge (C1) starts in year 0. This leads to a relatively sharp peak in the discharge in years 300–600.

As the Heinrich events are part of the internal variability of the model, there is no control run without the effect of Heinrich events. Therefore, we use a composite of the 400 years (40 years in the atmosphere) before the events (here referred to as pre-surge phase) as reference state. We define two further time windows for the analysis: the surge phase (years 300–600),
which covers the peak discharge, and the post-surge phase (years 1100–1400), which covers the minimal ice sheet volume at the end of the surge. During the surge phase, the climate is maximally affected by the freshwater discharge while the height of the ice sheet is slowly decreasing. In the post-surge phase, the ice discharge returns to its background level, while the ice sheet height loss reaches its maximum and its signal becomes dominant. As all averages are taken across the four events, they cover a total of 1600 years for the pre-surge phase, and 1200 years for the surge and post-surge phases.

## 3   Results

### 3.1   The climate in the pre-surge phase

The climate in the pre-surge phase is a mean of different times before the Last Glacial Maximum (Tab. 1). Therefore, it is representative of the cold phase of the last glacial, but not specifically of the Last Glacial Maximum, which is used in most steady-state glacial climate studies. The individual pre-surge states show only small deviations from their mean discussed here (Supplement Fig. 1), with global mean temperatures deviating from the ensemble-mean by less than 0.05 K, and the ice sheets mainly differing by the slow growth of the spurious ice sheet in eastern Siberia (Supplement Fig. 2). The global mean near-surface air temperature in the pre-surge phase is 3.2 K below the temperature in a pre-industrial experiment with the same model. The cooling is strongest over the ice sheets (Fig. 3). The North Atlantic Deepwater (NADW) is mainly formed south of Greenland and in the Nordic Seas (Fig. 5c). The NADW cell of the Atlantic Meridional Overturning Circulation (AMOC) has a maximum strength of 19 Sv at 30°N and reaches down to about 2900 m. For pre-industrial conditions, the modeled AMOC has the same depth and a strength of 15.8 Sv. The stronger AMOC under glacial conditions is a common feature in model simulations (Weber et al., 2007). Proxies are somewhat ambiguous with respect to the strength of the glacial AMOC (Klockmann et al., 2016), but clearly show a shoaling that is missing in our model. In our model, the AMOC strengthening most likely results from wind-stress changes induced by the larger ice sheets having a slightly stronger effect than the reduced greenhouse gas concentration (Klockmann et al., 2016).

The Laurentide ice sheet is split into a Cordilleran and a Hudson Bay part (Fig. 4). The Hudson Bay part is connected to the Greenland Ice Sheet, which has grown to fill the entire Greenlandic shelf. The connection between the Greenland and the Laurentide Ice Sheet during the glacial is well established, as is its widening (e. g. Lecavalier et al., 2014). The exact outlines of the LGM Greenland Ice Sheet are still under debate, with growing evidence for advances far onto the shelf (Arndt, 2018). The details in the model results should, however, be taken with a grain of caution, as the resolution of all components is rather low. Similarly, Iceland is covered in ice. The Fennoscandian Ice Sheet extends over large parts of Scandinavia as well as the Barents and Kara Sea shelves. In Eastern Siberia, a spurious ice sheet has formed (Section 2.2).

### 3.2   The ice dynamics

A large part of the ice discharge into the ocean is channeled in ice streams. Some of them are constantly active (Fig. 2), while most of them, such as the Hudson Strait Ice Stream (24 in Fig. 2), perform surge cycles alternating between active and inactive states (see video supplement). The underlying mechanism for the surges is related to the binge-purge cycles described by MacAyeal (1993). Similar cycles are also observed in a variety of mountain glaciers and have been related to the thermal and precipitation regime of these glaciers (Sevestre and Benn, 2015). In a warm / high precipitation regime, glaciers tend to have a continuously warm base and strong flow often with basal sliding. In cold / low precipitation regimes, glaciers are generally cold based without basal sliding. In an intermediate regime, neither state is stable, and many glaciers perform surge cycles.

The ice discharge at the mouth of Hudson Strait strongly varies (Fig. 1). When the Hudson Strait Ice Stream and the Ungava Bay Ice Stream (16 in Fig. 2) are quiescent, only 1 mSv of freshwater is discharged. Much of the time, the Ungava Bay Ice

stream delivers about 5 mSv. At intervals of about 5 000 years, the Hudson Strait Ice Stream surges for a period of about 1000–1500 years with a peak discharge of about 50 mSv (Fig. 1).

All surges start with low discharge rates, while the surge is propagating inland in Hudson Strait (Figs. 1, 2). Between years 150 and 250 of the composite, the surges in all events reach Hudson Bay (not shown). This causes the steep increase in the discharge signal, that we use to align the events. Around year 450, the northern part of Hudson Bay is surging in all four events. The exact shape of the surging area varies between the experiments. In A2 and B1 a second pulse in the discharge results from a delayed propagation of the surge into the southern part of Hudson Bay. In A1, this region is reached in the early stage of the surge. Since the propagation of the surge varies from event to event, so does the duration with a range of 860 years to 1450 years (Table 1). The surge drains ice from the Hudson Bay area and the surface elevation decreases by up to 900 m (Fig. 4). The strongest decrease in surface elevation is located in Hudson Strait. The losses tail off towards the interior of Hudson Bay. The eastern part of the Laurentide Ice Sheet reaches its minimum volume around year 1200 of the composite. The ice loss of 0.8 Mio Gt, from 16.2 Mio Gt (year 50) to 15.4 Mio Gt (year 1200), corresponds to a sea level rise of about 2.3 m. This is followed by a slow recovery of the ice sheet, which takes more than 3000 years (Fig. 1). All other ice sheets show no significant deviations from the background behavior with growth of the Eastern Siberian Ice sheet in ExA and onset of the deglaciation in the late recovery phases of events B1 and C1.

### 3.3 The ocean response

Our coupling scheme represents iceberg calving and sub-shelf melt as a freshwater flux with an associated negative latent heat flux to the ocean. Thus, when the ice from the Hudson Strait Ice Stream enters the ocean, sea ice forms in the ocean model. During the surge phase (years 300–600), most of the additional sea ice follows the Labrador Current out of the Labrador Sea and melts in the open North Atlantic (Fig. 6a). On its way, the sea ice shields the ocean from the atmosphere and thus contributes to a reduction in ocean heat release (Fig. 8), especially in winter, when the decrease in heat release reaches $70 \, \mathrm{W \, m^{-2}}$ in parts of the Labrador Basin (not shown) and the near surface air temperature decreases by up to 4 K (Fig. 5e for the annual mean). In summer, the increased sea-ice cover increases the albedo and reduces the net shortwave absorption with a maximum decrease of $20 \, \mathrm{W \, m^{-2}}$ at the mouth of Hudson Strait (not shown). When the sea ice melts, the surface water freshens (Fig. 5a) and the stratification stabilizes (see below). In the post-surge phase (years 1100–1400), the sea-ice concentration in the Labrador Sea partially returns to pre-surge levels. In the Irminger Basin, the sea-ice margin advances, and the sea-ice cover in the Nordic Seas grows (Fig. 6a). The increased ice cover matches with a weaker subpolar gyre (see below; Fig. 7a) and a following slight expansion of the East Greenland Current, indicated by a cooling and freshening at the eastern Margin of the current (Fig. 5b, d). In the Nordic Seas, the additional sea ice reduces the winter ocean heat release by up to $50 \, \mathrm{W \, m^{-2}}$ (not shown), exceeding the regional reductions during the surge-phase.

During the surge phase, the freshwater released from the melting ice takes a strongly zonal path, and spreads across the whole Atlantic between 40°N and 50°N (Fig. 5a). Most of it passes south of the deep water formation areas (Fig. 5a; Fig. 5c for the pre-surge phase mixed layer depth). Still, some of it reaches the deep water formation areas, and the sea surface salinity (SSS) in the areas with a time maximum winter mixed layer depth of more than 1000 m decreases by $0.35 \, \mathrm{g \, kg^{-1}}$. The mixed

layer depth in all deep water formation areas is reduced by up to 800 m (compare Fig. 5a, c) and the 300 year running mean of the maximum winter mixed layer volume in the Atlantic north of 40°N, the Nordic Seas and the Arctic Ocean shrinks from 6.4 Mio km$^3$ in the pre-surge phase to 4.7 Mio km$^3$ (−27%) during the surge phase. As a result, the NADW cell weakens from 18.3 Sv to 14.8 Sv (-19 %) in the same period (Fig. 8). The Arctic Ocean SSS also decreases (Fig. 5a). This is mostly due to a

4 mSv surge of the Amundsen Gulf Ice Stream (18 in Fig. 2), 10% of the Hudson Strait flux. The freshening in the Arctic Ocean is limited by the halocline depth (not shown), so the total volume affected is much smaller than in the case of the Hudson Strait surge. In the post-surge phase, the sea-surface salinity in the North Atlantic largely recovers, but remains below its pre-surge levels. The SSS in the deep water formation areas remains about 0.25 g kg$^{-1}$ below pre-surge levels. It reaches the pre-surge values after year 1500, when the mixed layer volume also recovers (not shown). The winter mixed layer depth in the central

North Atlantic and the Nordic Seas remains decreased, while it recovers off the coasts of Ireland and Scotland (Fig. 5b, d). The still reduced mixed layer depth could be the reason why the salinity remains decreased even after the termination of the surge. Less deep mixing can be caused by a freshwater lid, on the other hand, it also means that less mid-depth high salinity water is brought to the surface. The NADW cell has mostly recovered (Fig. 8b).

The increase in sea-ice cover and the reduction in the deep water formation caused by the ice discharge during the surge

phase lead to lower sea surface temperatures in the North Atlantic (Fig. 5c). In the Labrador Sea, and along the sea-ice margin, this cooling can be at least partly attributed to the increased sea-ice cover (Fig. 6a). The cooling is strongest in the open North Atlantic. With the slowdown of the meridional overturning circulation (Fig. 8b) less heat is brought into the subpolar gyre and the entire water column cools (not shown) and the heat release to the atmosphere is reduced on a large scale (Fig. 8c for the time series). In the post-surge phase, the meridional overturning circulation and the sea-surface temperatures largely recover. The

remaining cooling signal is strongest at the sea ice margin (Figs. 5d, 6b). Along the margin between subtropical and subpolar gyre, a slight northward shift of the gyre front leads to a surface warming (Fig. 5d) that contrasts with generally cool sea surface temperatures. This warming is more expressed at intermediate depth and reaches down to 1500 m (not shown).

During the surge phase, the low salinity anomaly caused by the melting ice is entrained in the NADW. The combination of freshening and cooling leads to a moderated reduction in the density, so the NADW formation remains active, albeit weakened.

Because of the NADW's reduced density, the Atlantic Antarctic Bottom Water (AABW) cell can expand slightly, decreasing the temperatures in the transition zone between NADW and AABW at a depth of about 3 km (not shown). The Atlantic subpolar gyre strengthens and reaches up to 55 Sv. This strengthening occurs despite a reduced wind driven component (Fig. 7, computed from the windstress curl by integrating the Sverdrup relation from east to west in each basin; Sverdrup, 1947). The strengthening of the gyre is caused by an increase in the cross-gyre density gradient (not shown). The convection brings

fresher, but also colder water down in the core of the subpolar gyre. The temperature effect in the core partially offsets the salinity effect. At the same time, the margins of the gyre freshen without significant temperature change, and the density contrast between core and margins of the gyre grows. In the post-surge phase, temperature and salinity begin to recover. The in-situ density in the core of the gyre remains reduced, while it recovers faster in the margins. In combination with a still reduced wind driven component, this weakens the gyre to 42 Sv. The gyre reaches its pre-surge strength around year 1500,

with strong multi-centennial variability that matches the wind stress forcing (Fig. 7).

## 3.4 The atmosphere response

Heinrich events affect the atmosphere in two ways, here called the direct and the indirect effect. During the surge-phase, the indirect effect via freshwater-release induced changes in sea-ice concentration, and ocean heat transport and release (see above, Fig. 8c) dominates. In the post-surge phase, the freshwater release ceases, and the direct effect takes over: The decreasing ice sheet height (Fig. 4, overlays in 5e,f) directly influences the large-scale circulation. This effect already appears in the surge phase and is strengthened in the post-surge phase. It leads to a northward expansion of the jet stream over the west Atlantic (Fig. 9). This brings cold air into Europe (Fig. 5) and reduces the wind-driven transport in the Atlantic subpolar gyre (see above, Fig. 7).

During the surge phase, reductions in near-surface air temperatures (Fig. 5e) can be observed along the path of the sea-ice out of the Labrador Sea, and over the southern part of the subpolar gyre, where the decreases in sea surface temperature are largest. The reduced ocean heat release (Fig. 8c) shows that the ocean is causing this cooling, which then is spread across the high latitudes by the winds. Due to the smaller ice sheet, the temperatures in the Hudson Bay area rise (see below). In the post-surge phase (Fig. 5f), the cooling is strongest along the sea-ice margin (Fig. 6b), where the sea-surface temperature changes are amplified by the effects of the increased ice cover. In these areas the ocean heat loss still is reduced, while over the open North Atlantic the picture is less clear with areas of increased and decreased ocean heat release. Here the effects of the changes in jet stream and ocean circulation intermingle. The warming over Hudson bay is strengthened (see below).

Due to the reduced ocean heat transport and the consequential cooling of the North Atlantic sector during the surge phase, (Fig. 5c–f), the evaporation decreases by 10% (North Atlantic and all connected basins north of 45°N, not shown). Over Europe, this results in a decrease in precipitation by 4–5% (Figs. 10b, 11c) that spreads far into Siberia. A northwest-southeast dipole pattern in the precipitation over the tropical Atlantic indicates a southeastward shift of the intertropical convergence zone (ITCZ). This pattern extends into Africa, with a southward shift of the tropical rain belt leading to a dipole pattern with drying in the Sahel Zone and an increase in the precipitation over southern Africa. In the post-surge phase, the evaporation over the North Atlantic partially recovers (not shown) and so does the precipitation in Europe (Fig. 10c, 11d). A slight large-scale reduction in evaporation remains in the tropical Atlantic, and the Sahel region (not shown). The dipole precipitation pattern in the Atlantic ITCZ turns into a monopole with a slight precipitation increase (Fig. 10c). The dipole pattern over Africa is slightly weakened but remains intact.

In the Hudson Bay area, the surface air temperature rises throughout the surge. This is in part due to the lapse-rate effect of the reduced surface elevation resulting from the surge. However, a closer look reveals that in both, the surge phase and the post-surge phase, the warming is more zonal and centered further to the south than expected from the lapse-rate effect alone (Fig. 5e, f). In the following, we describe the changes in the Hudson Bay area during the post-surge phase, as the basic behavior during the surge-phase is identical, just with a lower amplitude. We find an increase in total heat transport convergence in the southern Hudson Bay area by about $3 \, \mathrm{W \, m^{-2}}$ (not shown). The eastern half of the area of depression experiences an increase in cloud cover (Fig. 11a, b). It is associated with an increase in surface net longwave radiation by $5\text{-}6 \, \mathrm{W \, m^{-2}}$, while the net shortwave radiation remains constant (not shown). The increase in cloud cover brings an increase in precipitation, locally

reaching $50\,\mathrm{mm\,yr^{-1}}$ (+50 %, Fig. 11). The increased precipitation decreases the mass loss of the ice sheet during the surge and accelerates its regrowth.

## 4    Discussion

The surges in our ice sheet model are based on an internal oscillation of the ice sheet, as described analytically by MacAyeal
(1993) and first modeled in a three-dimensional ice sheet model by Calov et al. (2002). While the occurrence and properties of these oscillations are highly model-dependent (Calov et al., 2010), they are a robust feature of our ice sheet model in coupled as well as stand-alone setups (not shown). Such surge cycles have also been observed by other groups using PISM (Van Pelt and Oerlemans, 2012; Feldmann and Levermann, 2017), and cause the surges in GLIMMER in Roberts et al. (2014b, 2016). Such self-sustained surge cycles are known to occur in thousands of mountain glaciers (e. g. Sevestre and Benn, 2015), and
their details are linked to the intricacies of sub-glacial hydrology. Robel et al. (2013) provides a recent zero-dimensional model of ice-hydrology interactions in the context of Heinrich events. However, self-sustained oscillations of the ice sheet are not the only possible explanation for Heinrich events (see below).

For modeling surges in our model with the Shallow Shelf Equation stress balance controlling the sliding, a horizontal resolution that properly resolves Hudson Strait proved necessary. In our simulations, the Hudson Strait Ice Steam was constantly
active at 40 km model resolution (not shown); doubling the resolution to 20 km yielded the surge cycles. The sliding coefficient (equation 1) controlling the basal sliding is higher than in Calov et al. (2002, 2010), partly because about 60% of the gravitational driving stress are compensated for by membrane stresses (not shown) and thus not available for driving the sliding. While the value of the friction coefficient is substantially higher than values commonly obtained for Antarctica or Greenland, the geological history of the Hudson Bay area vastly differs from that of Antarctica or Greenland. This might explain for dif-
ferent basal conditions. The study of Roberts et al. (2016) showed only modest differences in the Heinrich events when varying the basal sliding coefficient between the value used in our study and that of Calov et al. (2002), thus, while the coefficient is high in our setup, this is not critical for the validity of our results

In models, the surge intensity and period are model dependent and affected by the surface temperature, the surface mass balance, as well as by the basal friction law (Calov et al., 2010). The slightly too short period of 5 kyr in our model instead of
the 7 kyr derived from proxy records can be due to any combination of these factors. Since this kind of experiment takes a very long time on the supercomputer, we could not fine-tune the surge interval. However, the peak discharge rate of about 50 mSv agrees with the result obtained by Roberts et al. (2014a, b, 2016). Our surges with a duration of 0.9–1.5 kyr are shorter than the 3 kyr surges in their experiments, and with a cycle duration of 5 kyr, they are more frequent compared to 11 kyr cycles in their experiments. The surges in our study show an average loss of $0.96\,\mathrm{Mio\,km^3}$ of ice, as compared to $2.5\,\mathrm{Mio\,km^3}$ in their
experiments. The surges show a very similar peak discharge rate. This is most likely set by the geometry of the Hudson Strait limiting the flow. Despite this, even the surges occurring at the same time show different discharge histories. ExB and ExC are initialized shortly before the surge, and the only difference between the setups is the routing of the Nile river. Accordingly, the ice sheets are very similar at the beginning of the surge (Fig. 1b, Supplement Fig. 2). Then, however, their evolution diverges

due to the extreme nonlinearity of the processes involved in the surge with switching between fast sliding and non-sliding basal conditions. The similarity in basic shape and peak discharge as well as the differences between individual realizations resulting from the non-linearities are in perfect agreement with idealized studies (Calov et al., 2010; Soucek and Martinec, 2011) as well as Roberts et al. (2016).

The pure self-oscillating system does not explain for the occurrence of Heinrich events in Dansgaard-Oeschger stadials. Sub-surface ocean warming observed in proxies has been identified as a possible trigger (Moros et al., 2002) with mechanisms ranging from ocean-induced melt triggering a rapid retreat of the ice stream (Bassis et al., 2017), via a repeatedly collapsing Labrador Sea ice shelf controlling the flow speed of the Hudson Strait Ice Stream (Álvarez-Solas et al., 2011; Alvarez-Solas et al., 2013) to the whole Heinrich event being the break-up of an ice shelf (Hulbe et al., 2004). As the glacial ocean circulation is very stable in MPI-ESM (Klockmann et al., 2018), we cannot study the relationship between the modeled Heinrich events and Dansgaard-Oeschger cycles. Thus, this study is not meant to provide an answer on the exact mechanics behind the ice sheet collapses and their occurrence during stadials, but as an investigation of the consequences of such ice-sheet collapses on the climate system.

The comparison to climate proxy reconstructions is somewhat complicated, as Heinrich events generally fall into the Dansgaard-Oeschger stadials which are missing in our model. Thus, we cannot investigate the combined effects of precursory climate changes and the ice sheet collapse, but focus on the climate response to the pure ice sheet collapse. In this, we are in accordance with all previous Heinrich event modeling studies. Climate proxies, however, contain a combined signal of the stadial with its northern hemisphere cooling and the additional effect of the ice discharge.

During the surge phase, the freshwater discharge is the dominant forcing in our model. This can be seen from the timing of the strong cooling and reduction in ocean heat release over the North Atlantic that strictly follows the freshwater input and SSS changes (Figs. 1, 5, 8). The decrease in the NADW cell strength by 19 % is on the weak end of freshwater hosing studies (Stouffer et al., 2006; Otto-Bliesner and Brady, 2010; Kageyama et al., 2013). This is of little surprise, as the freshwater perturbation from the surging ice sheet is weaker than the forcings prescribed in most of these studies, and the decrease of the NADW cell strength is highly correlated to the strength of the freshwater forcing (e. g. Maier-Reimer and Mikolajewicz, 1989; Stouffer et al., 2006). Furthermore, the modeled discharge into the Labrador sea allows a large fraction of the freshwater to escape into the subtropical gyre, while in most hosing experiments, the freshwater is spread more efficiently over the deepwater formation areas.

During the surge phase, we observe a reduction in temperature and SSS across the North Atlantic (Fig. 5a,c), which is confirmed by proxies from the eastern side of the North Atlantic (e. g. Heinrich, 1988; Bard et al., 2000). During the post-surge phase, a slight ocean cooling remains, but the SSS largely recovers (Fig. 5b,d). On land, pollen records from around the Mediterranean and western Europe (e. g. Tzedakis et al., 2004; Fletcher and Sánchez Goñi, 2008; Fletcher et al., 2010) show a correlation between the Dansgaard-Oescheger cycles and tree types and cover. The pollen indicate relatively cold, arid climates during stadials and relatively warm, humid climates during the interstadials. Among the stadials, the Heinrich Stadials are the coldest and most arid. In the model, we observe colder, and, over large parts of Europe also drier conditions during the surge and post-surge phases (Fig. 10), matching the trend in the proxies. In the surge phase, this is a result of the aforementioned

AMOC decrease, in the post-surge phase, the lowered Laurentide Ice Sheet allows the jet stream to expand northwards, and to take a more northerly path across the Atlantic. This is the main mechanism of the topography change experiments of Roberts et al. (2014b), and consistent with simulations by Merz et al. (2015), who showed that changes in the glacial topography triggered anomalies in the stationary wave activity in their atmosphere-land-only model. These anomalies lead to a shift in the eddy-driven jet stream. In sensitivity experiments with glacial and present-day ice sheet configurations under similar climate forcing, they found that the presence of the Laurentide Ice Sheet was the dominant driver for a southward shift and acceleration of the jet stream. The dependence of the jet stream path on ice sheet height is also known from other simulations (Ullman et al., 2014). The slightly different timing of the modeled climate changes with land changes lasting longer than ocean changes calls for high-resolution proxies with reliable cross-dating across the land/ocean interface.

In the tropics, the most prominent change in the model and in proxies is a southward shift of the ITCZ (e. g. Arz et al., 1998; Hessler et al., 2010). Over Africa, we observe this shift in the surge as well as in the post-surge phase. Over the Atlantic, it is a southward shift during the surge phase and a pure increase in the post-surge phase. The shift is also observed in freshwater hosing experiments, where the surface elevation effect generally is not represented. In our experiments as well as in the fresh-water hosing studies, the freshening reduces the Atlantic heat transport and thus causes a dipole anomaly in the sea surface temperatures with cooling in the North (Fig. 5c) and a slight warming in the South (not shown), resulting in a southeast shift of the ITCZ (e. g. Stouffer et al., 2006).

Around the Gulf of Mexico, our model yields mixed results in the comparison with proxy data. For all Heinrich events, Grimm et al. (2006) find an increase in pine tree pollen and indication for an increased lake level in Lake Tulane (central Florida). Our model simulates an increase in precipitation north of Florida, and drying in Florida and the Caribbean during the surge phase (Fig. 10b) and large-scale drying in the post-surge phase (Fig. 10c). However, this is a mismatch by only one grid cell in the surge phase. Rühlemann et al. (2004) find a rapid warming of 1–3 K for intermediate depth (1299 resp. 426 m) waters at 17 ka in benthic foraminifera from sediment cores from the Carribean Sill and Angolan coast. Our model shows an sub-surface warming extending from 60 to 400 m at the Caribbean sill reaching 2.4 K in the surge phase and 0.7 K in the post-surge phase, and no change at the Angolan coast (not shown).

In the Greenland ice cores, the Heinrich stadials do not appear colder, but generally longer than the normal Dansgaard-Oeschger stadials (e. g. Bond et al., 1993). Our model simulates an increase in the sea-ice cover (Fig. 6), and colder (Fig. 5) and drier conditions during both the surge- and post-surge phase. These effects could plausibly prolong a pre-existing stadial in the ice core records.

The EPICA Community Members (2006) present data from the Byrd (80°S, 110.5°W), EPICA Dronning Moud Land (EDML, 75°S, 0°E), and EPICA Dome C (75°S, 123.3°E) ice cores. They show warming trends during the Dansgaard-Oeschger Stadials reaching values between 0.5 and 3 K, with longer and thus stronger warmings during Heinrich stadials. This warming is generally ascribed to the bipolar see-saw effect of an AMOC weakening (Stocker and Johnsen, 2003). Our model simulates a general cooling trend over most of Antarctica in both phases, with the exception of an 0.1 K warming at EDML during the surge phase. The bipolar see-saw effect could possibly be obtained in a model with a less stable ocean circulation, or with a stronger ice discharge, both leading to a stronger AMOC reduction.

The different phases in our experiments correspond to the different experiments of Roberts et al. (2014b). They obtain the southward ITCZ shift, that we observe in the surge phase, only for a high freshwater release, while the experiment with pure topography changes shows the increase in precipitation that we obtain in the post-surge phase. However, not all changes agree between our simulation and that of Roberts et al. (2014b). In our model, there is a west-east gradient in the precipitation change across Hudson Bay with increases on the eastern side (Figs. 10, 11). This contrasts with the results of Roberts et al. (2014b), where the precipitation over Hudson Bay decreases while the strongest increases are evident southeast of the bay. One factor contributing to such different behavior might be the different ice sheet shapes. We simulate a two-part Laurentide Ice Sheet, Roberts et al. (2014b) simulate one contiguous Laurentide Ice Sheet with the main dome in the southwest corner. As more coupled ice sheet–climate models are developed, further coupled studies will help distinguishing between model-specific and more robust effects.

Jongma et al. (2013) model HE1 as a 300 year 0.235 Sv discharge under LGM conditions. When including iceberg freshwater and latent heat fluxes, they simulate the strongest surface freshening in the frontal zone of the Newfoundland Basin, with a strong decrease of the freshening towards the east. They show that the feedback of the latent heat flux from iceberg melt on the ocean drastically reduces the iceberg melt in the Labrador sea as compared to modeling the freshwater flux only, and that it leads to an increased formation of sea ice. By modeling the calving as sea ice, we over-represent this effect. The melt is much more concentrated on the sea-ice edge, as the sea ice is not able to cross the open North Atlantic. A sensitivity test with our model shows that the inclusion of the latent heat effect of ice discharge increases the ice cover and the cooling in the Labrador Sea, but has minimal consequences outside of this area (Supplement Fig. 3). All in all, the surface freshening is more zonal during the main discharge phase in our experiments than in Jongma et al. (2013). Icebergs are not strictly tied to the ocean circulation and can cross gyre fronts more easily. This allows for a wider spread of the freshwater signal. Modeling the latent heat effect without modeling icebergs still is an improvement over the traditional treatment of completely neglecting it.

## 5 Conclusions

By modeling Heinrich events as MacAyeal–style binge-purge cycles in a coupled ice sheet–climate model system, we are able to observe a two-step response of the climate system: first to the freshwater discharge, and later to the elevation decrease over the Laurentide Ice Sheet. The freshwater discharge strengthens the stratification of the North Atlantic surface waters. The stronger stratification reduces the deep water formation. This weakens the North Atlantic meridional overturning circulation and reduces the heat loss to the atmosphere. The cooling of the North Atlantic affects the North-South sea surface temperature gradient and thus shifts the Atlantic intertropical convergence zone southeastward. This behavior is also known from freshwater hosing experiments. Furthermore, the cooling over the North Atlantic reduces the evaporation and thus precipitation in this region and downwind in Eurasia. In contrast to freshwater hosing experiments, we also see the effects of the reduced surface elevation on the atmospheric circulation. The surface elevation effects reach their maximum when the freshwater discharge ceases and the elevation of the Laurentide Ice Sheet is at its minumum. The lower elevation allows for a northward expansion of the jet stream. This weakens the subpolar gyre and reduces the heat transport into (northern) Europe. Furthermore the

lowered surface elevation allows for more precipitation over the Hudson Bay area and thus speeds up the regrowth of the Laurentide Ice Sheet.

Modeling ice sheets in fully coupled ice sheet–climate model systems allows to study the full interactions between ice sheets and the climate. Only by using an ice sheet model, we could obtain a physically plausible and self-consistent set of freshwater fluxes and ice sheet surface changes that led to the observed two-stage behavior, explaining the previous results from hosing experiments as well as elevation change experiments.

## 6 Code availability

MPI-ESM is available under the Software License Agreement version 2 after acceptance of a license (https://www.mpimet.mpg.de/en/science/models/license/; MPI, 2018). PISM is available under the GNU General Public License at http://www.pism-docs.org.

## 7 Data availability

The model output can be obtained from the DKRZ World Data Center for Climate (WDCC) at https://cera-www.dkrz.de/WDCC/ui/cerasearch/entry?acronym=DKRZ_lta_989, (WDC Climate, 2018).

*Author contributions.* F. Z. and U. M. developed the setup. U. M. performed the experiments. F. Z. analyzed the data and wrote the manuscript. All authors discussed the analysis and manuscript.

*Competing interests.* U. M. is a member of the editorial board of the journal but his accesses to this manuscript have been frozen following the journal policy. The authors declare that they have no other conflicts of interest.

*Acknowledgements.* This work was supported by German Federal Ministry of Education and Research (BMBF) as Research for Sustainability initiative (FONA); through the project PalMod (FKZ: 01LP1502A and 01LP1504C). All simulations were performed at the German Climate Computing Center (DKRZ). The authors thank William Roberts and Thomas Kleinen for comments which helped to improve the manuscript. Further thanks go to André Paul for kindly handling the manuscript as well as to Jorge Álvarez-Solas and an anonymous reviewer for improving the manuscript with their helpful comments and suggestions.

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

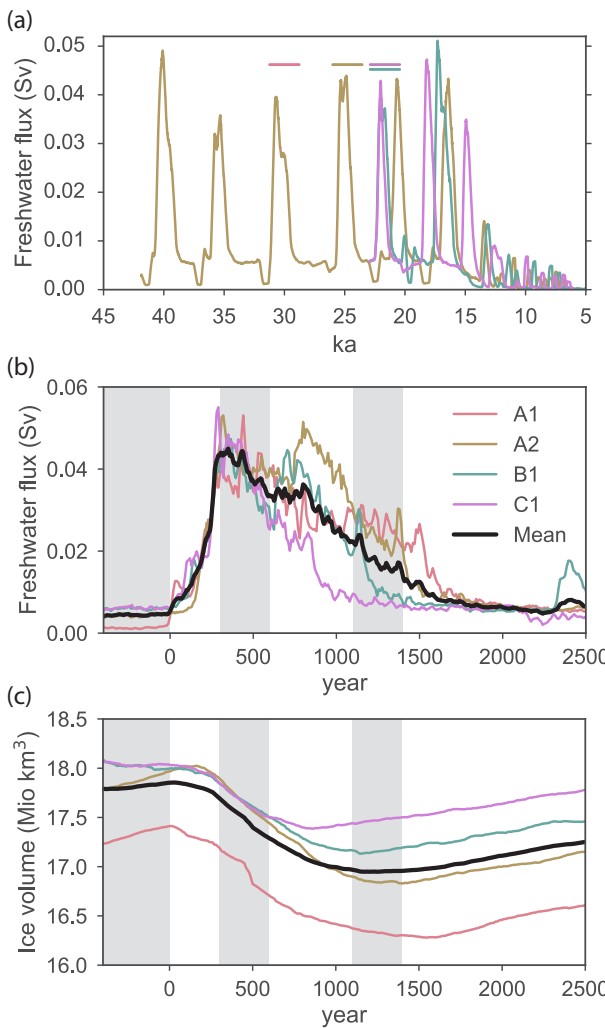

**Figure 1.** (a), (b) Hudson Strait Ice Stream discharge, (c) ice volume of the eastern part of the Laurentide Ice Sheet. (a) is averaged over 300 years, (b) over 30 years, (c) is plotted from instantaneous values. Line colors in (a) correspond to Heinrich events A2, B1, and C1, horizontal bars mark the events selected for the composite analysis in the colors used throughout the time series analysis. The vertical gray bars in (b,c) mark the pre-surge (years −400 to 0), surge (years 300 to 600), and post-surge (years 1100 to 1400) phases (Section 2.3). 1 Mio km$^3$ of ice corresponds to about 2.5 m of sea level.

| ID | Experiment | Surge start composite year | Surge Start years BP | Surge End years BP | Duration years | Max flux mSv | Mean flux mSv |
|---|---|---|---|---|---|---|---|
| A1 | ExA | 110 | 31060 | 29610 | 1450 | 53 | 29 |
| A2 | ExA | 70 | 25680 | 24350 | 1330 | 53 | 34 |
| B1 | ExB | 35 | 22420 | 21270 | 1150 | 47 | 30 |
| C1 | ExC | 0 | 22420 | 21560 | 860 | 55 | 28 |

**Table 1.** Events investigated. The surge start is defined by the surge branching off the Ungava Bay ice stream and propagating towards Hudson Bay. Its time is given in years BP and as the year in the composite. The end of the surge is defined as the time when there is no more surging in Hudson Strait. The maximum flux is defined as the maximum of 30-year running means, matching Fig. 1b.

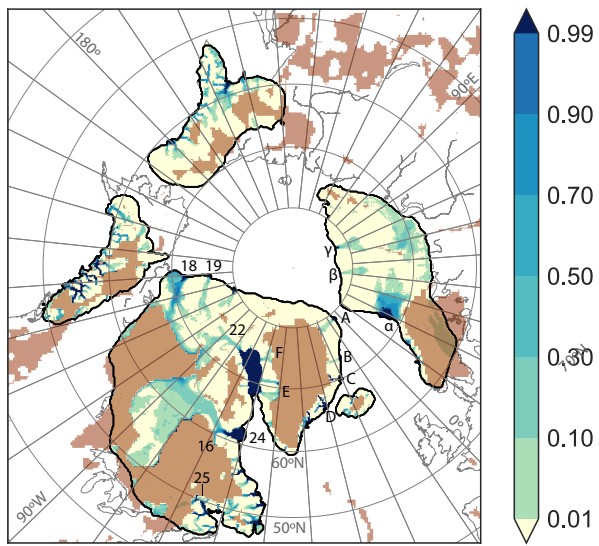

**Figure 2.** Fraction of time that a grid cell in the glacial part of ExA (41 ka–15.6 ka) is sliding or has floating ice with a basal velocity exceeding $1 \, \mathrm{m \, yr^{-1}}$. Only the time when the grid cell is ice covered is taken into consideration; therefore, ice shelves are considered to be constantly sliding. Only areas with a time-mean ice thickness in excess of 10 m are displayed. Brown marks areas where the ice is not permitted to slide due to the lack of sediments in the reconstruction of Laske and Masters (1997). Numbers match the ice stream numbering in Stokes and Tarasov (2010), and are as follows: (16) Ungava Bay, (18) Amundsen Gulf, (19) M'Clure Strait, (22) Lancaster Sound, (24) Hudson Strait, (25) Laurentian. Letters refer to present-day location names of modeled Greenland ice streams: (A) Northeast Greenland Ice Stream, (B) Kejser Franz Joseph Fjord, (C) Scoresby Sund, (D) Kangerdlugssuaq, (E) Jakobshaven Isbrae, (F) Kong Oscar Glacier. Greek letters $(\alpha, \beta, \gamma)$ mark Barents Shelf ice streams. In the Baltic Sea, the ice sheet retreats at the start of the experiment, so there temporarily is an ice shelf. Stereographic projection around 71°N, 44°W with grid lines at multiples of 10°.

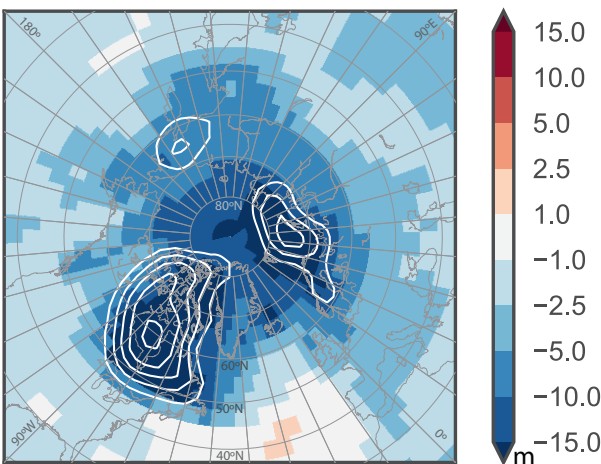

**Figure 3.** Temperature difference between the pre-surge phase and a pre-industrial experiment, overlay: topography increase from pre-industrial to pre-surge glacial state (mostly ice sheets) as seen by ECHAM5 at multiples of 500 m. For a definition of the pre-surge phase see Fig. 1 and Section 2.3. Polar stereographic projection with grid lines at multiples of $10°$.

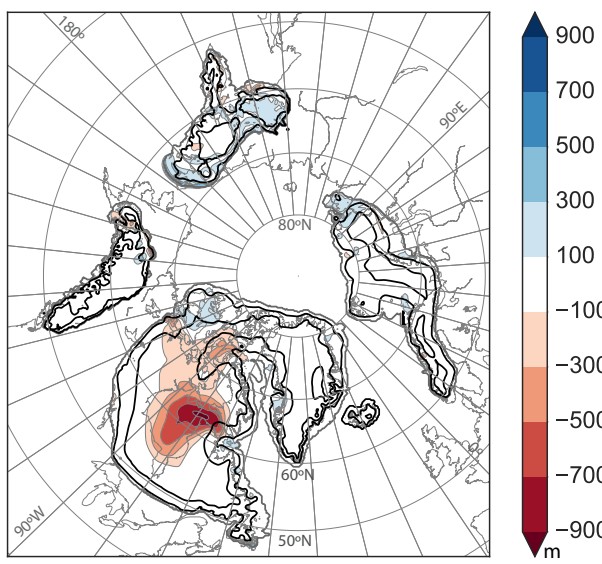

**Figure 4.** Ice surface elevation change in composite year 1250 compared with composite year $-50$. Overlays mark ice sheet outline (gray) and surface elevation (black) at 1000, 2000, 3000 m in the pre-surge phase. The Laurentide ice sheet barely misses 3000 m. The peak of Greenland slightly exceeds this elevation. Stereographic projection around $71°$N, $44°$W with grid lines at multiples of $10°$.

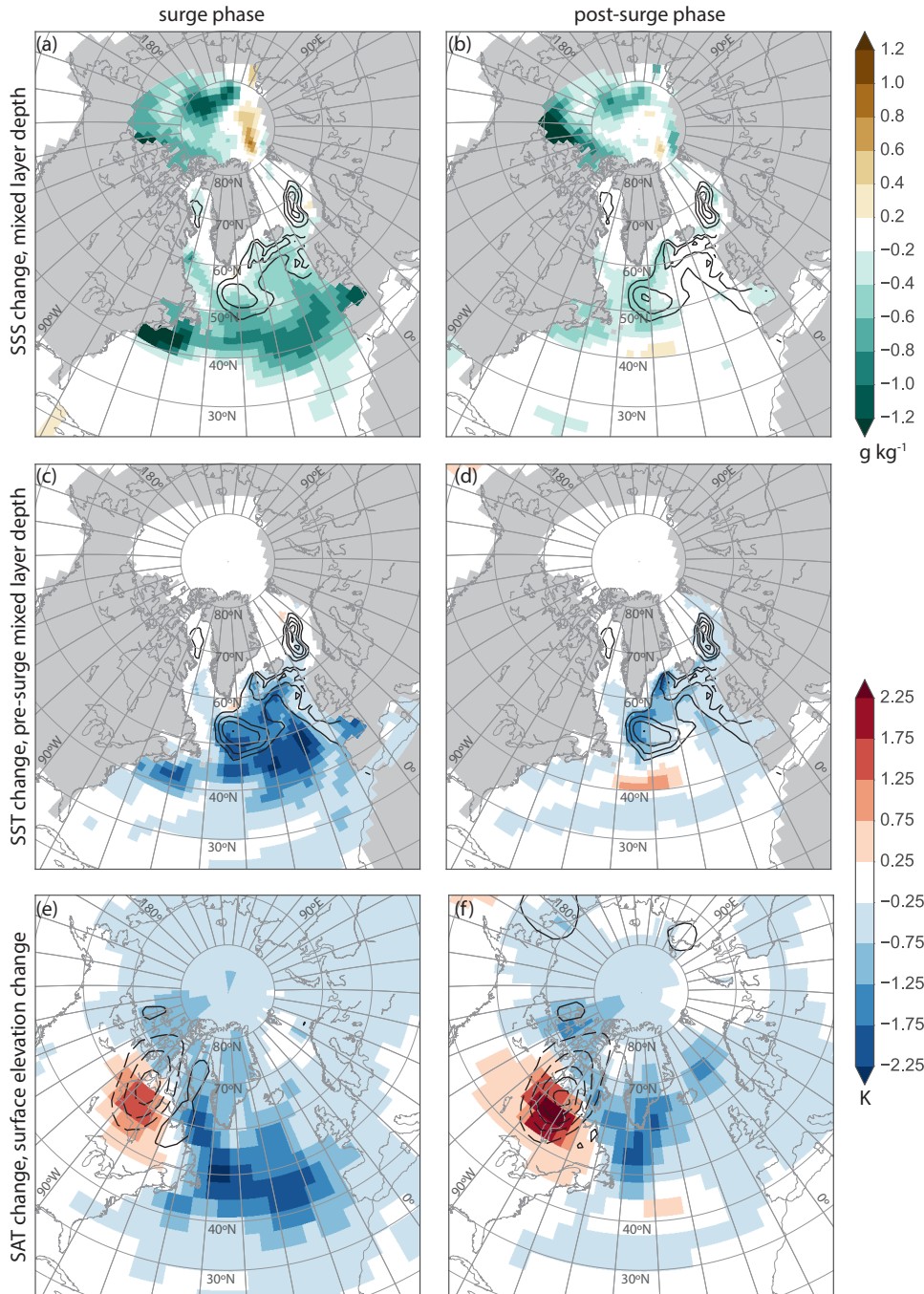

**Figure 5.** (a,b) Sea surface salinity, (c,d) sea surface temperature, and (e,f) surface air temperature changes compared to the pre-surge state during (a,c,e) the surge phase and (b,d,f) the post-surge phase. The overlays show (a,b) maximum mixed layered depth during the respective phases of the event at multiples of 500 m (maximum of the 300-year climatology), (c,d) maximum mixed layer depth in the pre-surge phase (same scale), and (e,f) elevation changes in the respective phases of the event vs. the pre-surge phase (at multiples of 100 m, dashed lines mark negative values, solid lines positive values). For a definition of the phases see Fig. 1 and Section 2.3. Stereographic projection around $67^\circ$N, $44^\circ$W with grid lines at multiples of $10^\circ$.

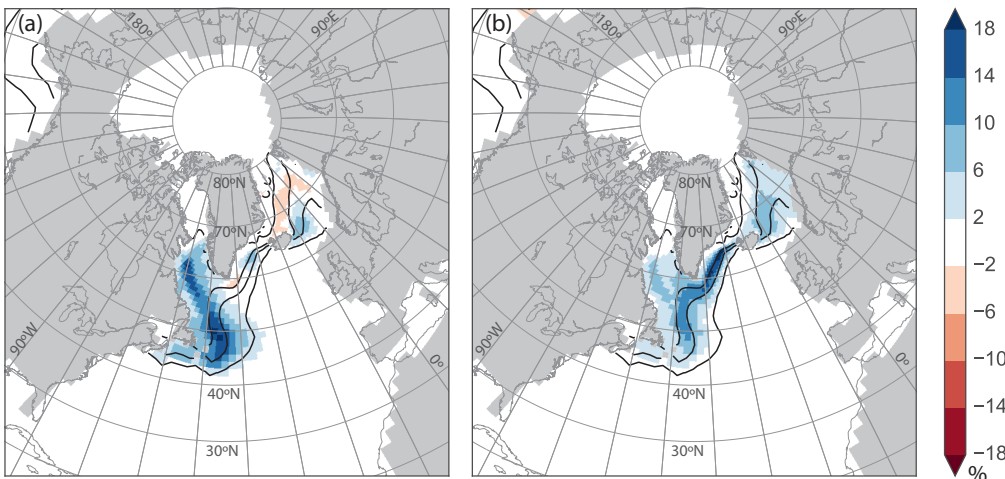

**Figure 6.** Sea-ice concentration change compared to the pre-surge phase for (a) surge phase, and (b) post-surge phase. Overlays mark sea-ice concentration in the pre-surge phase on levels of 1, 15, 50, and 95%. For a definition of the phases see Fig. 1 and Section 2.3. Stereographic projection around 67°N, 44°W with grid lines at multiples of 10°.

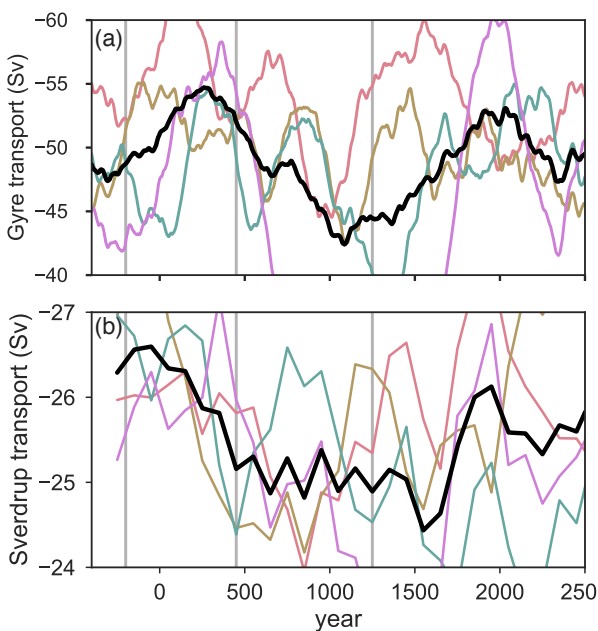

**Figure 7.** Strength of the North Atlantic subpolar gyre and the windstress contribution (computed from the windstress curl by integrating the Sverdrup relation from east to west in each basin, Sverdrup, 1947). (a) 300 year running mean, (b) 300 year block means. Colors as in Fig. 1. Vertical lines mark the centers of the intervals selected for the analysis. As this figure is using 300 year averaging, they indicate the average values corresponding to the time windows used in the analysis. Note the different scales for gyre and Sverdrup transport.

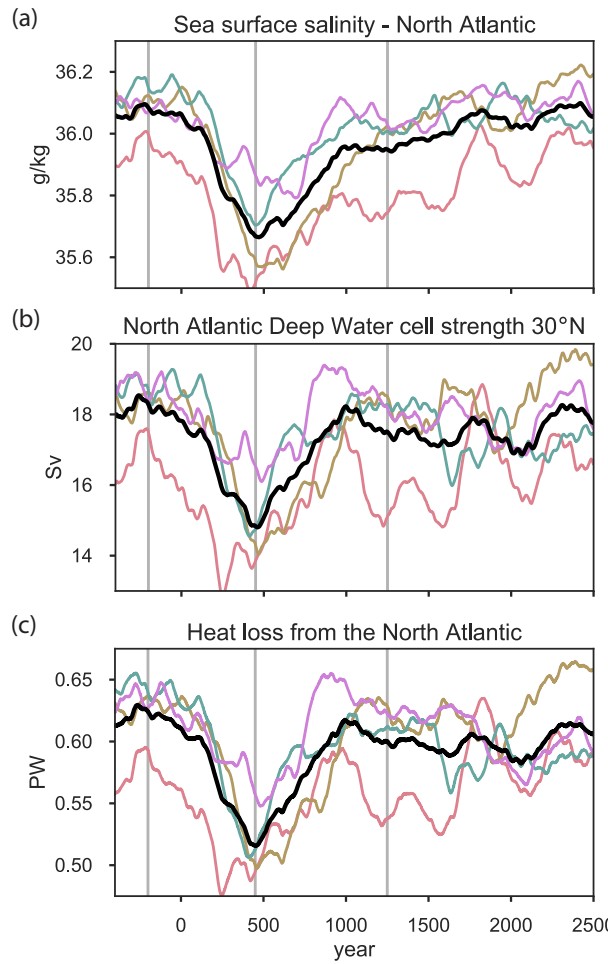

**Figure 8.** (a) North Atlantic Sea surface salinity, (b) NADW cell strength at 30°N and (c) North Atlantic heat loss. 300 year running means. Colors as in Fig. 1, vertical lines as in Fig. 7.

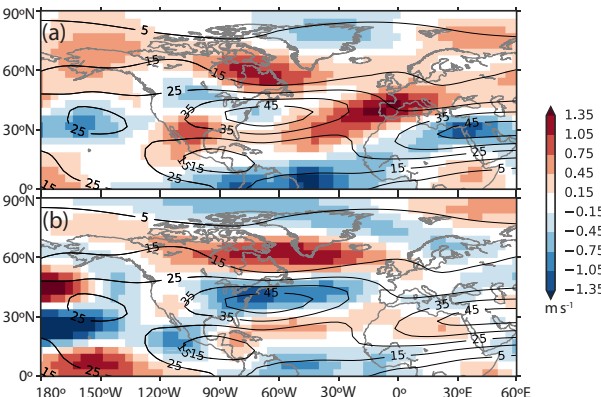

**Figure 9.** DJF 200 hPa zonal wind speed anomalies compared to the pre-surge phase. Overlays show zonal wind speeds in the pre-surge phase in ms$^{-1}$. Dashed lines mark negative values. (a) surge phase, (b) post-surge phase. Note the projection with focus on the North Atlantic. For a definition of the phases see Fig. 1 and Section 2.3.

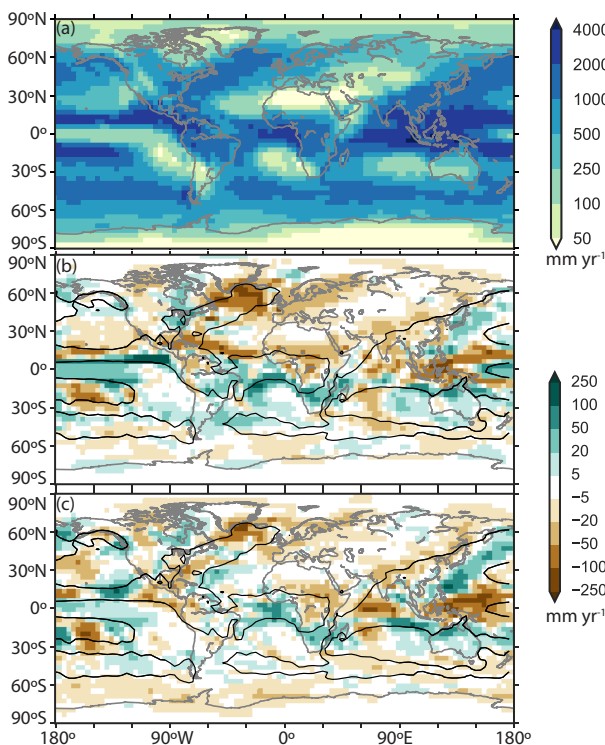

**Figure 10.** (a) annual mean precipitation in the pre-surge phase, and precipitation changes in (b) the surge phase and (c) the post-surge phase with pre-surge 1 m yr$^{-1}$ isoline overlain to allow for an easier identification of the ITCZ. For a definition of the phases see Fig. 1 and Section 2.3.

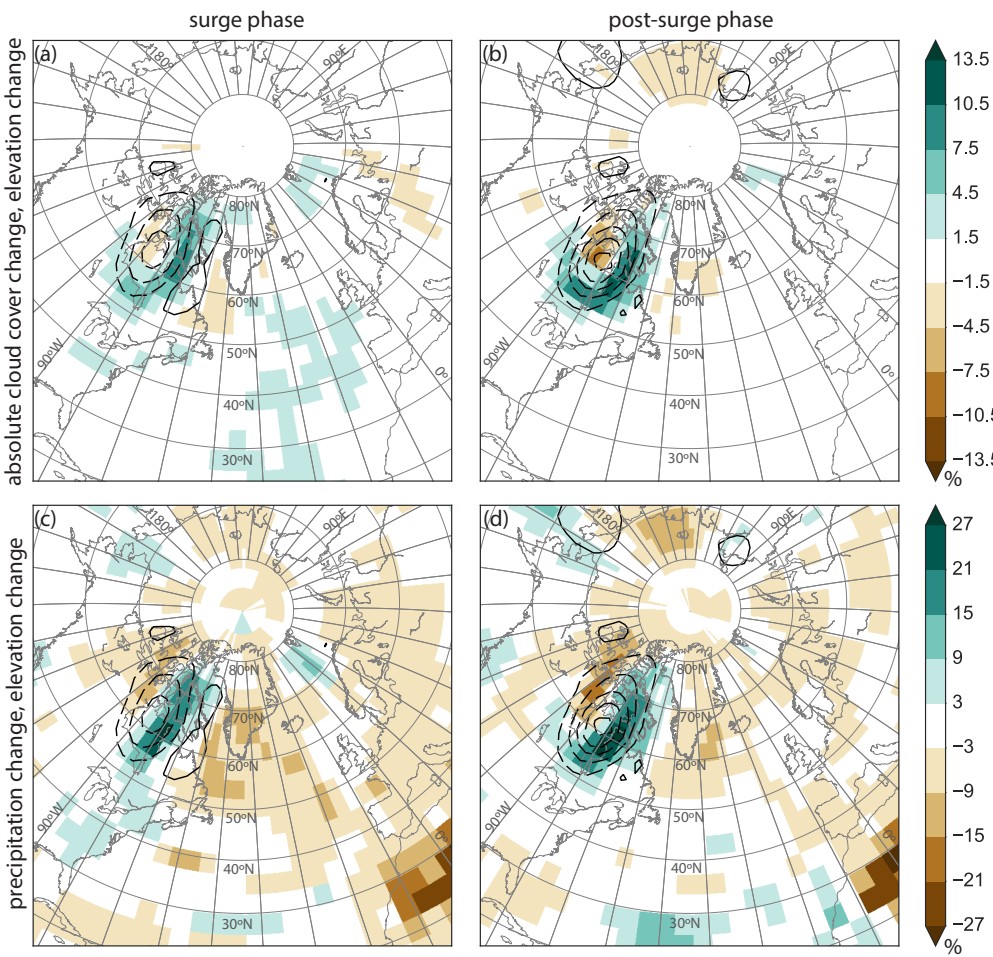

**Figure 11.** (a,b) Total cloud cover changes, and (c,d) relative changes in precipitation. (a,c) surge phase, and (b,d) post-surge phase. Isolines mark the ice sheet elevation change at multiples of 100 m. Dashed lines mark negative values. All changes are relative to the pre-surge phase. For a definition of the phases see Fig. 1 and Section 2.3. Stereographic projection around 67°N, 44°W with grid lines at multiples of 10°.