# Peer review of "Heinrich events show two-stage climate response in transient glacial simulations"

_Climate of the Past, 2018_

## Referee Comment (RC1) · Anonymous Referee #1 · 28 Jun 2018

The study of Ziemen et al. studies H-events with a coupled climate-ice sheet model. The model finds internal oscillations with a recurrence time of approximately 5kyr, in reasonable agreement with the observed climate record. This study is one of very few coupled climate model studies of H-events, given that the timescales involved are so large that it has, until now, not been feasible to run these types of experiments.

The authors report a two-state response to the internal H-events in the mode. The first state is characterised by freshwater release from land based ice sheets (dominated by the Hudson Bay area), while the second phase is characterised by a reduction in ice-sheet elevation (mostly Hudson Bay) and changes in atmospheric circulation and precipitation.

The paper is well written and is highly novel as it is one of very few, if any, coupled

climate-ice sheet models of this detail studying the H-events of the last glacial period. However, there are a few concerns which should be considered before publication in climate of the past.

GENERAL COMMENTS:

The results of the study are clearly novel and of great importance to our understanding of H-events and instabilities of the ice sheets surrounding the Atlantic Ocean during the last glacial. The model includes several assumptions which should be better discussed. In particular, the coupling scheme includes period-synchronous 1:10 coupling. Still it is not clear from reading the methods section what this means, and how this choice of coupling method might impact the results. Please elaborate on this.

Another key assumption in the model is that the freshwater input is associated with a negative heat input when ice enters the ocean. However, it is not clear from the manuscript how this impacts the results. More detail should be given to this point. How important is it for the response observed?

The authors choose to study H-events in a transient glacial climate simulation which makes the analysis challenging. It is not clear how this change in boundary conditions impacts the results, and it is not clear why the authors choose a composite of several transient runs in their results description. This should be better explained/discussed in the revised manuscript. Note, however, that including a transient glacial climate is also of interest ast it could give clues as to how the H-events change with time. It would be of benefit to the study if such an assessment were to be included.

Note also that the duration and amount of freshwater for the different experiments are surprisingly similar (See table 1). The reasons for this should be discussed.

The glacial state is not well defined in the manuscript. More detail should be given to the difference between the background climate states at the time of the H-events simulated in the model.

And finally, one of my major concerns with this study is the lack of comparison of the results with proxy data. Given that there is a wealth of data highlighting changes in ocean, and atmosphere climate during the glacial and across H-events the model results should be discussed in relation to these. In particular, as the study clearly states its relevance for studying H-events of the past.

SPECIFIC COMMENTS:

Line 11, page 3: chose either 2D or 3D. Would assume it is both which are relevant here, but only 2D fields can be shown in the paper.

Line 14, page 3: it is stated that PDD method is applied. Please give more details of how this is implemented in the model.

Line 7, page 4: there is a reference to spurious ice over Siberia. What is this and what does it mean? Give more detail with reference to figure e.g. . ..

Line 8, page 5: The pre-surge AMOC has a strength of 19Sv, whereas the PI AMOC is 15.8 Sv (see Section 3.1). Why is there a difference, and why is the glacial run AMOC stronger? How does this compare with other model studies and with data?

Line 11, page 5: it is stated that the Laurentide is connected to Greenland at the glacial in the model, including the entire Greenland shelf. This is an interesting result which should be further discussed. Is this expected from data, are there similar findings by other studies?

Line 15, page 5: it is stated that the Hudson strait Ice stream has a cycle in surging, whereas other ice streams are constant. Explain why. It is clear from the results that the Hudson Bay system is special, with binge-purge type oscillations. Why is this only the case here? Are there other similar systems?

Line 30, page 6: it is stated the convection depth changes. However, this is not clear form figure 5. It would be beneficial to add more details in terms of change in e.g. mixed layer depth or similar.

[Figure]

Line 18, page 7: it is stated that the freshwater is drawn down by deep water formation. What is this based on? If this is true, please show it in a figure or similar. Puzzling that freshwater can be drawn down given its low salinity.

Line 15, page 10: it is stated that the jet stream changes. Why is this? Explain.

---

## Referee Comment (RC2) · J. Alvarez-Solas (Referee) · 5 Oct 2018

Review of "Heinrich events show two-stage climate response in transient glacial simulations" by Ziemen et al, 2018.

I am sorry for the delay in finishing this review and I apologize to the authors for the derived inconvenients.

The work of Ziemen et al, analyses the effects of internally-produced ice surges of the Laurentide ice sheet on the Northern Hemisphere climate around to the LGM. It does so in a fully coupled (asynchronous) ice-sheet / climate framework. Such a modelling framework has an inherent merit and it is of sufficient interest to make this contribution worth of being published. The paper nicely analyses the effects of a freshwater injection on the North Atlantic behavior and, because being fully coupled, it also describes the impacts of such an oceanic change on the ice sheet at the same time as the show the impacts of the lowering height of the Laurentide on the Northern Hemisphere climate. This represents an important contribution under a more realistic framework when compared to the classic hosing experiments done with climate models alone. Hosing experiments have been useful in order to understand what are the consequences of a reduction of North Atlantic density (by means of prescribed freshwater fluxes) on the rest of the climate system. The current work has the advantage of providing such a flux in a physically-based maner within the context of a coupled ice-climate system.

That being said, I think there are two main assumptions in the current manuscript that need to be discussed:

1) The authors somehow assume in the discussion section that Heinrich events in the real world arise as self-sustained cyclic surges of the Hudson strait ice stream.

2) Following the logic of 1), the observed climatic changes during Heinrich stadials are interpreted to be merely the consequence of the above mentioned surges (page 1, lines 2 and 19; page 9 lines 26-32)

Regarding 1): There are in the literature relatively recent papers not cited here defending that the triggering of Heinrich events lies on an oceanic forcing (Bassis et al, Nature 2017; Alvarez-Solas et al, PNAS 2013), rather than on a binge-purge-like mechanism. Furthermore, making the ice streams of the Laurentide ice sheet oscillate in a 3D thermomechanical model is subjected to technical nuances in the way the basal movement of the ice is treated. In particular, I see one choice (inherited from the experimental setup described in Ziemen et al, 2014), that deserves further attention or at least a caveat in the manuscript. I.e. C = 1 m/yr/Pa is a very high (and likely unrealistic) value for a linear sliding law because:

a) In Calov et al, 2002 (the first to show binge-purge-like oscillations in a 3D thermomechanical ice sheet), the chosen value of C was 0.1 m/yr/Pa (10 times smaller than

in the current manuscript). And then, as sensitivity tests, the effects of considering even smaller values of that parameter (until 0.01 m/yr/Pa; 100 times smaller) were discussed.

b) One could wonder what would the magnitudes of the simulated velocities in present-day Antarctica following a linear sliding law ($U_b = C \tau_b$) with $C = 1$ m/yr/Pa be. Taking a look at Morlighem et al, 2013, for example, and using their inferred basal stresses ($\tau_b$), the reader would be surprised by the resulting velocities of the antarctic ice streams, ranging from 20 to more than 100 km/yr. In fact, this approach can also be followed inversely. I.e. given the observed velocities and the inferred basal stresses, one can deduce what the values of the sliding parameter would be. So, dividing the observed velocities ($U_b$) by the basal stresses ($\tau_b$) in Morliguem et al, 2013, the resulting median value of C is 0.02 m/yr/Pa. This is 50 times smaller than the one used for producing the cycling Laurentide surges in the current manuscript.

I guess (because ice velocities are not shown here) that, thanks to including the non-local SSA solution and its propagation of longitudinal stresses (as opposed to the propagation of the surface slope under the local SIA solution), the ice flow is stabilized and therefore such extremely high velocities are prevented to appear in the model. Additionally, I am aware that the realism of the cyclic Hudson strait ice streams surges produced here (called Heinrich events in the manuscript) is not the main focus of the paper. Thus, producing new ice-sheet simulations with smaller values of C is probably not necessary for the current paper. Nonetheless, what seems necessary is to acknowledge that the robustness of the glaciological mechanisms producing self-sustained Laurentide ice surges is (at least) under debate and that therefore simply calling these oscillations Heinrich events could be premature.

With respect to 2): Heinrich events always occur during cold phases of the observed millennial variability in Greenland (during stadials) but not for every stadial. A convenient explanation of the phenomenon would simply be (and has been) that the ice surges from the Laurentide trigger (or facilitate) the shift into stadials. However, more

evidence is growing pointing to the fact that icebergs in the North Atlantic appear in sediment cores significantly after the cooling of the stadials is already observed (Barker et al, 2015). This implies that the iceberg discharges from Laurentide surges are not the responsible of the observed cold phases neither during stadials nor during Heinrich stadials. This is a bit in contradiction with what the current manuscript suggests (see for example page 1 in the introduction: "... iceberg armadas spread detritus from the Hudson Strait area across the North Atlantic seafloor and caused large-scale climate changes."). The value of the simulations shown and analysed in the manuscript is not affected by what is exposed above. However, acknowledging that the chain of causes and effects explaining the observed climate features during HEs might be not as simple as previously thought (the ice sheet surges, circulation and density drop and thus the ocean cools) would, in my opinion, improve the paper.

Specific comments: Section 2.2 is not very clear to me: What is the purpose of having 3 different realisations of the model forced with the same boundary conditions and internal parameters? Is the spin-up procedure shared between exps B and C? Do they have the same initial conditions as well? If yes, why are they producing the surges at different times?

Jorge Alvarez-Solas

---

## Author Comment (AC1) · 16 Nov 2018

We thank the anonymous reviewer for the positive evaluation of our manuscript and for the helpful comments and suggestions. Below please find a point-to-point response to the comments.

GENERAL COMMENTS:

The results of the study are clearly novel and of great importance to our understanding of H-events and instabilities of the ice sheets surrounding the Atlantic Ocean during the last glacial. The model includes several assumptions which should be better discussed. In particular, the coupling scheme includes period-synchronous 1:10 coupling. Still it is not clear from reading the methods section what this means, and how this choice of coupling method might impact the results. Please elaborate on this.

We expanded the paragraph:

*In all simulations used for the analysis, a configuration was used where atmosphere and ocean are coupled with a periodic-synchronous 1:10 coupling (Voss and Sausen, 1996; Mikolajewicz et al., 2007a). The motivation for this is that the atmosphere has no long-term memory (this resides in ocean and ice sheets), but consumes more than 90% of the CPU time of the coupled model system. Periodic-synchronous coupling means that the atmosphere model is only run for one out of ten years. This reduces the computational expense drastically and speeds up the simulations by a factor of three (wallclock time). After each fully coupled year, the ocean is forced for nine years with atmospheric fields, that are obtained by cycling through the previous five fully coupled years. These settings turned out to be a good compromise between having a sufficiently large archive (of atmospheric forcing) to adequately represent the inter-annual variability of the atmospheric forcing and minimizing the delay in the forcing from a large archive. A too small archive leads to a large model drift in the ocean-only phases and thus corrupts the climate of the coupled model, a large archive introduces a large delay. Whereas this coupling technique has only minor effect on the simulated climate response to long-term changes, this technique should not be used for short-term changes or the analysis of short-term variability. With the settings used her the periodic-synchronous coupling introduces a lag of up to 50 years in the atmosphere-ocean system, the average lag of about 25 years is less than 10% of the 300-year averaging window used throughout most of the analysis. Furthermore, most of the changes in the ice sheets driving the atmosphere-ocean system occur on a longer time scale, so the lag can be neglected in the analysis. Parallel to the climate model, the ice sheet model is run for ten years, so ice sheets and ocean are on the same time scale. In the analysis of atmosphere and ocean fields, only the fully coupled years are used. These are also used to obtain the fields for forcing the ice sheet model. Thus, when we speak of a 300 year mean of the ocean 2D fields, only 30 years of data with a 10-year spacing are used. This should yield comparable results to the full data as long as no signals with periods of exact multiples of 10 years are involved (aliasing).*

Another key assumption in the model is that the freshwater input is associated with a negative heat input when ice enters the ocean. However, it is not clear from the manuscript how this impacts the results. More detail should be given to this point. How important is it for the response observed?

We only apply the effect to freshwater released by direct ice-ocean interactions (shelf melt, calving), not to surface runoff. The effect mainly consists of an increase in the ice cover in Hudson Bay. We still consider modeling this effect. superior to neglecting it. We provide a few comparison plots of freshwater hosing experiments with and without the latent heat flux as supplement Figure 2 and added

*A sensitivity test with our model showed that the inclusion of the latent heat effect of ice discharge increases the ice cover and the cooling in the Labrador Sea, but has minimal consequences outside of this area (Supplement Fig. 2).*

The authors choose to study H-events in a transient glacial climate simulation which makes the analysis challenging. It is not clear how this change in boundary conditions impacts the results, and it is not clear why

the authors choose a composite of several transient runs in their results description. This should be better explained/discussed in the revised manuscript. Note, however, that including a transient glacial climate is also of interest as it could give clues as to how the H-events change with time. It would be of benefit to the study if such an assessment were to be included.

We added sentences in the experiments section and in the description of the composite analysis:

Experiments:

*We chose these simulations as technically quasi-identical subset from various simulations that were performed when working on a model that is able to simulate the last deglaciation. As the simulations consumed considerable resources, we refrained from performing a dedicated ensemble, but made use of the available data.*

Composite analysis:

*The mechanisms related to the surging of the ice sheet are highly non-linear, leading to variability between individual realizations of the modeled events even under quasi-identical conditions (Soucek and Martinec, 2011). This variability is further amplified by feedbacks in the fully coupled ice sheet–climate model. To reduce the influence of variability and thus obtain more robust results, we perform all further analysis on a composite of all four events.*

Note also that the duration and amount of freshwater for the different experiments are surprisingly similar (See table 1). The reasons for this should be discussed.

We expanded the discussion:
The surges show a very similar peak discharge rate. This is most likely set by the geometry of the Hudson strait limiting the flow. Despite this, even the surges occurring at the same time surprisingly dissimilar. ExB and ExC are initialized shortly before the surge, and the only difference between the setups is the routing of the Nile river. Accordingly, the ice sheets are very similar the beginning of the surge (Fig. 1b). Then, however, their evolution diverges due to the extreme non-linearity of the processes involved in the surge with switching between fast sliding and non-sliding basal conditions. The similarity in basic shape and peak discharge as well as the differences between individual realizations resulting from the non-linearities are in perfect agreement with idealized studies (Calov et al., 2010; Soucek and Martinec, 2011) as well as Roberts et al. (2016).

The glacial state is not well defined in the manuscript. More detail should be given to the difference between the background climate states at the time of the H-events simulated in the model.

We added:

*The individual pre-surge states show only small deviations from their mean discussed here (Supplement Fig. 1)*

And finally, one of my major concerns with this study is the lack of comparison of the results with proxy data. Given that there is a wealth of data highlighting changes in ocean, and atmosphere climate during the glacial and across H-events the model results should be discussed in relation to these. In particular, as the study clearly states its relevance for studying H-events of the past.

We re-wrote large parts of the discussion and included comparisons with proxy data.

[revised manuscript text omitted]

SPECIFIC COMMENTS:

Line 11, page 3: chose either 2D or 3D. Would assume it is both which are relevant here, but only 2D fields can be shown in the paper.

We chose 2D.

Line 14, page 3: it is stated that PDD method is applied. Please give more details of how this is implemented in the model.

We expanded to:

*The surface mass balance is computed using downscaled precipitation and temperatures in a Positive Degree Day (PDD, Reeh, 1991) scheme in the ice sheet model. The PDD scheme employs the Calov–Greve integral method (Calov and Greve, 2005) to compute PDDs from monthly mean temperatures and standard deviations. The PDDs are then converted to snow and ice melt. As in Ziemen et al. (2014), the temperature standard deviation for the PDD scheme is computed from 6- hourly atmosphere model output. Extending this method, a minimum sub-monthly standard deviation of 4 K is prescribed. This prevents the standard deviation from falling too low in areas, where ECHAM5 simulates melt and limits the surface temperature to $0°C$.*

Line 7, page 4: there is a reference to spurious ice over Siberia. What is this and what does it mean? Give more detail with reference to figure e.g. . . .

We added

*(Fig. 8 of Ziemen et al. (2014), its re-emergence can be seen in Fig. 2)*

Line 8, page 5: The pre-surge AMOC has a strength of 19Sv, whereas the PI AMOC is 15.8 Sv (see Section 3.1). Why is there a difference, and why is the glacial run AMOC stronger? How does this compare with other model studies and with data?

We added:

The stronger AMOC under glacial conditions is a common feature in model simulations (Weber et al., 2007). Proxies are somewhat ambiguous with respect to the strength of the glacial AMOC (Klockmann et al., 2016), but clearly show a shoaling that is missing in our model. In our model, the AMOC strengthening most likely results from wind-stress changes induced by the larger ice sheets having a slightly stronger effect than the reduced greenhouse gas concentration (Klockmann et al., 2016).

Line 11, page 5: it is stated that the Laurentide is connected to Greenland at the glacial in the model, including the entire Greenland shelf. This is an interesting result which should be further discussed. Is this expected from data, are there similar findings by other studies?

We added:

*The connection between the Greenland and the Laurentide Ice Sheet during the glacial is well established, as is its widening (e. g. Lecavalier et al., 2014). The exact outlines of the LGM Greenland Ice Sheet are still under debate with growing evidence for advances far onto the shelf (Arndt, 2018). The details in the model results should, however, be taken with a grain of caution, as the resolution of all components is rather low.*

Line 15, page 5: it is stated that the Hudson strait Ice stream has a cycle in surging, whereas other ice streams are constant. Explain why. It is clear from the results that the Hudson Bay system is special, with binge-purge type oscillations. Why is this only the case here? Are there other similar systems?

We expanded to clarify:

*A large part of the ice discharge into the ocean is channeled in ice streams. Some of them are constantly active (Fig. 2), while most of them, such as the Hudson Strait Ice Stream (24 in Fig. 2), perform surge cycles alternating between active and inactive states (see video supplement). The underlying mechanism for the surges is related to the binge-purge cycles described by MacAyeal (1993). Similar cycles are also observed in a variety of mountain glaciers and have been related to the thermal and precipitation regime of these glaciers (Sevestre and Benn, 2015). In a warm / high precipitation regime, glaciers tend to have a continuously warm base and strong flow often with basal sliding. In cold / low precipitation regimes, glaciers are generally cold based without basal sliding. In an intermediate regime, neither state is be stable, and glaciers perform surge cycles.*

Line 30, page 6: it is stated the convection depth changes. However, this is not clear form figure 5. It would be beneficial to add more details in terms of change in e.g. mixed layer depth or similar.

We clarified:

*The mixed layer depth in all deep water formation areas is reduced by up to 800 m (compare Fig. 5a, c)*

Line 18, page 7: it is stated that the freshwater is drawn down by deep water formation. What is this based on? If this is true, please show it in a figure or similar. Puzzling that freshwater can be drawn down given its low salinity.

We clarified:

*During the surge phase, the low salinity anomaly caused by the melting ice is entrained in the NADW. The combination of freshening and cooling leads to a moderated reduction in the density, so the NADW formation remains active, albeit weakened.*

Line 15, page 10: it is stated that the jet stream changes. Why is this? Explain.

We expanded:

[revised manuscript text omitted]

---

## Author Comment (AC2) · 16 Nov 2018

Dear Jorge, thank you very much for the kind comments and helpful suggestions. Below please find a point-to-point response to the comments.

I am sorry for the delay in finishing this review and I apologize to the authors for the derived inconvenience. Thank you very much for taking your time to review this manuscript.

The work of Ziemen et al, analyses the effects of internally-produced ice surges of the Laurentide ice sheet on the Northern Hemisphere climate around to the LGM. It does so in a fully coupled (asynchronous) ice-sheet / climate framework. Such a modelling framework has an inherent merit and it is of sufficient interest to make this contribution worth of being published. The paper nicely analyses the effects of a freshwater injection on the North Atlantic behavior and, because being fully coupled, it also describes the impacts of such an oceanic change on the ice sheet at the same time as the show the impacts of the lowering height of the Laurentide on the Northern Hemisphere climate. This represents an important contribution under a more realistic framework when compared to the classic hosing experiments done with climate models alone. Hosing experiments have been useful in order to understand what are the consequences of a reduction of North Atlantic density (by means of prescribed freshwater fluxes) on the rest of the climate system. The current work has the advantage of providing such a flux in a physically-based manner within the context of a coupled ice-climate system.

That being said, I think there are two main assumptions in the current manuscript that need to be discussed:
1) The authors somehow assume in the discussion section that Heinrich events in the real world arise as self-sustained cyclic surges of the Hudson strait ice stream.
2) Following the logic of 1), the observed climatic changes during Heinrich stadials are interpreted to be merely the consequence of the above mentioned surges (page 1, lines 2 and 19; page 9 lines 26-32)

We agree that we were too superficial in discussing these aspects, and therefore expanded on their discussion in the introduction (largely rewritten) and the discussion. We consider the causation of Heinrich events by an internal oscillation as the most plausible mechanism proposed so far. We are aware that an additional mechanism is needed to explain the phasing and consider the triggering by sub-surface ocean warming described in your works and the related work of Hulbe et al. (2004) by Bassis et al. (2017) as the best explanations so far. Calov et al. (2002) have shown the possibility of having a self-sustained oscillation together with ocean triggering. We rephrased the introduction and discussion to clarify that we are aware of the discussion regarding the mechanisms of Heinrich events, and the importance of precursory climate changes (largely the effects of Dansgaard-Oeschger stadials) in the analysis of the climatic signals. We will detail on these changes below.

Regarding 1): There are in the literature relatively recent papers not cited here defending that the triggering of Heinrich events lies on an oceanic forcing (Bassis et al, Nature 2017; Alvarez-Solas et al, PNAS 2013), rather than on a binge-purge-like mechanism. Furthermore, making the ice streams of the Laurentide ice sheet oscillate in a 3D thermomechanical model is subjected to technical nuances in the way the basal movement of the ice is treated. In particular, I see one choice (inherited from the experimental setup described in Ziemen et al, 2014), that deserves further attention or at least a caveat in the manuscript. I.e. C = 1 m/yr/Pa is a very high (and likely unrealistic) value for a linear sliding law because:
a) In Calov et al, 2002 (the first to show binge-purge-like oscillations in a 3D thermomechanical ice sheet), the chosen value of C was 0.1 m/yr/Pa (10 times smaller than
in the current manuscript). And then, as sensitivity tests, the effects of considering even smaller values of that parameter (until 0.01 m/yr/Pa; 100 times smaller) were discussed.

We added:
*Also the friction coefficient controlling the basal sliding is lower than in Calov et al. (2002, 2010), partly because about 60% of the driving stress are compensated for by membrane stresses (not shown). While the value of the friction coefficient is substantially lower than values commonly obtained for Antarctica or Greenland, the geological history of the Hudson Bay area vastly differs from that of Antarctica or Greenland. This might explain for different basal conditions.*

b) One could wonder what would the magnitudes of the simulated velocities in present-day Antarctica following a linear sliding law ($U\_b = C \tau\_b$) with C = 1 m/yr/Pa be. Taking a look at Morlighem et al, 2013, for example, and using their inferred basal stresses ($\tau\_b$), the reader would be surprised by the resulting

velocities of the antarctic ice streams, ranging from 20 to more than 100 km/yr. In fact, this approach can also be followed inversely. I.e. given the observed velocities and the inferred basal stresses, one can deduce what the values of the sliding parameter would be. So, dividing the observed velocities ($U\_b$) by the basal stresses ($tau\_b$) in Morlighem et al, 2013, the resulting median value of C is 0.02 m/yr/Pa. This is 50 times smaller than the one used for producing the cycling Laurentide surges in the current manuscript.

I guess (because ice velocities are not shown here) that, thanks to including the non-local SSA solution and its propagation of longitudinal stresses (as opposed to the propagation of the surface slope under the local SIA solution), the ice flow is stabilized and therefore such extremely high velocities are prevented to appear in the model. Additionally, I am aware that the realism of the cyclic Hudson strait ice streams surges produced here (called Heinrich events in the manuscript) is not the main focus of the paper. Thus, producing new ice-sheet simulations with smaller values of C is probably not necessary for the current paper. Nonetheless, what seems necessary is to acknowledge that the robustness of the glaciological mechanisms producing self-sustained Laurentide ice surges is (at least) under debate and that therefore simply calling these oscillations Heinrich events could be premature.

We added a paragraph to clarify this:
*The pure self-oscillating system does not explain for the occurrence of Heinrich events in Dansgaard-Oeschger stadials. Sub-surface warming ocean warming observed in proxies has been identified as a possible trigger (Moros et al., 2002) with mechanisms ranging from ocean-induced melt triggering a rapid retreat of the ice stream (Bassis et al., 2017), via repeatedly collapsing ice shelf controlling the flow speed of the Hudson Strait Ice Stream (Álvarez-Solas et al., 2011; Alvarez-Solas et al., 2013) to the whole Heinrich event being the break-up of an ice shelf (Hulbe et al., 2004 ). As the glacial ocean circulation is very stable in MPI-ESM (Klockmann et al., 2018), we cannot study the relationship between the modeled Heinrich events and Dansgaard-Oeschger cycles. Thus, this study is not meant to provide an answer on the exact mechanics behind the ice sheet collapses, but as an investigation of the consequences of an ice-sheet collapse on the climate system.*

With respect to 2): Heinrich events always occur during cold phases of the observed millennial variability in Greenland (during stadials) but not for every stadial. A convenient explanation of the phenomenon would simply be (and has been) that the ice surges from the Laurentide trigger (or facilitate) the shift into stadials. However, more evidence is growing pointing to the fact that icebergs in the North Atlantic appear in sediment cores significantly after the cooling of the stadials is already observed (Barker et al, 2015). This implies that the iceberg discharges from Laurentide surges are not the responsible of the observed cold phases neither during stadials nor during Heinrich stadials. This is a bit in contradiction with what the current manuscript suggests (see for example page 1 in the introduction: "... iceberg armadas spread detritus from the Hudson Strait area across the North Atlantic seafloor and caused large-scale climate changes."). The value of the simulations shown and analysed in the manuscript is not affected by what is exposed above. However, acknowledging that the chain of causes and effects explaining the observed climate features during HEs might be not as simple as previously thought (the ice sheet surges, circulation and density drop and thus the ocean cools) would, in my opinion, improve the paper.

(see above) We added a paragraph to clarify this:
*The pure self-oscillating system does not explain for the occurrence of Heinrich events in Dansgaard-Oeschger stadials. Sub-surface warming ocean warming observed in proxies has been identified as a possible trigger (Moros et al., 2002) with mechanisms ranging from ocean-induced melt triggering a rapid retreat of the ice stream (Bassis et al., 2017), via repeatedly collapsing ice shelf controlling the flow speed of the Hudson Strait Ice Stream (Álvarez-Solas et al., 2011;Alvarez-Solas et al., 2013) to the whole Heinrich event being the break-up of an ice shelf (Hulbe et al., 2004). As the glacial ocean circulation is very stable in MPI-ESM (Klockmann et al., 2018), we cannot study the relationship between the modeled Heinrich events and Dansgaard-Oeschger cycles. Thus, this study is not meant to provide an answer on the exact mechanics behind the ice sheet collapses, but as an investigation of the consequences of an ice-sheet collapse on the climate system.*

Specific comments: Section 2.2 is not very clear to me: What is the purpose of having 3 different realisations of the model forced with the same boundary conditions and internal parameters? Is the spin-up procedure shared between exps B and C? Do they have the same initial conditions as well? If yes, why are they producing the surges at different times?

We added paragraphs describing the reasons for the choice of experiments and their differences.
Introduction:
Experiments:
*We chose these simulations as technically quasi-identical subset from various simulations that were performed when working on a model that is able to simulate the last deglaciation. As the simulations consumed considerable resources, we refrained from performing a dedicated ensemble, but made use of the available data.*

Composite analysis:
*The mechanisms related to the surging of the ice sheet are highly non-linear, leading to variability between individual realizations of the modeled events even under quasi-identical conditions (Soucek and Martinec, 2011). This variability is further amplified by feedbacks in the fully coupled ice sheet–climate model. To reduce the influence of variability and thus obtain more robust results, we perform all further analysis on a composite of all four events.*

We expanded the Discussion:
*The surges show a very similar peak discharge rate. This is most likely set by the geometry of the Hudson strait limiting the flow. Despite this, they are surprisingly dissimilar. ExB and ExC are intialized shortly before the surge, and are virtually identical until the beginning of the surge (Fig. 1b)). Then, however, their evolution diverges due to the extreme nonlinearity of the processes involved in the surge with switching between fast sliding and non-sliding basal conditions. The similarity in basic shape and peak discharge as well as the differences between individual realizations resulting from the non-linearities are in perfect agreement with idealized studies (Calov et al., 2010; Soucek and Martinec, 2011) as well as Roberts et al. (2016).*

References:

Alvarez-Solas, J., Robinson, A., Montoya, M., and Ritz, C.: Iceberg discharges of the last glacial period driven by oceanic circulation changes, Proc Natl Acad Sci U S A, 110, 16350-16354, 10.1073/pnas.1306622110, 2013.
Álvarez-Solas, J., Montoya, M., Ritz, C., Ramstein, G., Charbit, S., Dumas, C., Nisancioglu, K., Dokken, T., and Ganopolski, A.: Heinrich event 1: an example of dynamical ice-sheet reaction to oceanic changes, Clim. Past, 7, 1297-1306, 2011.
Bassis, J. N., Petersen, S. V., and Cathles, L. M.: Heinrich events triggered by ocean forcing and modulated by isostatic adjustment, Nature, 542, 332-334, 2017.
Calov, R., Ganopolski, A., Petoukhov, V., Claussen, M., and Greve, R.: Large-scale instabilities of the Laurentide ice sheet simulated in a fully coupled climate-system model, Geophys. Res. Lett., 29, 69--61----69--64-69, 2002.
Calov, R., Greve, R., Abe-Ouchi, A., Bueler, E., Huybrechts, P., Johnson, J. V., Pattyn, F., Pollard, D., Ritz, C., Saito, F., and Tarasov, L.: Results from the Ice-Sheet Model Intercomparison Project–Heinrich Event INtercOmparison (ISMIP HEINO), J. Glaciol., 56, 371-383, 2010.
Hulbe, C. L., MacAyeal, D. R., Denton, G. H., Kleman, J., and Lowell, T. V.: Catastrophic ice shelf breakup as the source of Heinrich event icebergs, Paleoceanography, 19, PA1004-----PA1004, 2004.
Klockmann, M., Mikolajewicz, U., and Marotzke, J.: Two AMOC States in Response to Decreasing Greenhouse Gas Concentrations in the Coupled Climate Model MPI-ESM, Journal of Climate, 31, 7969-7984, 10.1175/jcli-d-17-0859.1, 2018.
Moros, M., Kuijpers, A., Snowball, I., Lassen, S., Bäckström, D., Gingele, F., and McManus, J.: Were glacial iceberg surges in the North Atlantic triggered by climatic warming?, Marine Geology, 192, 393-417, 2002.
Soucek, O., and Martinec, Z.: ISMIP-HEINO experiment revisited: effect of higher-order approximation and sensitivity study, J. Glaciol., 57, 1158-1170, 2011.
Ullman, D. J., LeGrande, A. N., Carlson, A. E., Anslow, F. S., and Licciardi, J. M.: Assessing the impact of Laurentide Ice Sheet topography on glacial climate, Clim. Past, 10, 487-507, 2014.